# Metallic Structures Based on Zinc Oxide Film for Enzyme Biorecognition

**DOI:** 10.3390/mi13111997

**Published:** 2022-11-17

**Authors:** Nicoleta Iftimie, Rozina Steigmann, Dagmar Faktorova, Adriana Savin

**Affiliations:** 1NDT Department, National Institute of Research and Development for Technical Physics, 700050 Iasi, Romania; 2Department of Design and Special Technology, Faculty of Special Technology, Trenčín University of Alexander Dubček in Trenčín, 911 01 Trenčín, Slovakia

**Keywords:** zinc oxide nanoparticles, enzymes biorecognition, biomaterials, gratings mesostructures, multifunctional nanosystem

## Abstract

Two structures (Ag/ZnO/ITI/glass: #1 sample and Ag/ZnO/SiO_2_/Si: #2 sample) are investigated, on the one hand, from the point of view of the formation of evanescent waves in the gratings of metal strips on the structures when the incident TEz wave in the radio frequency range is used. The simulation of the formation of evanescent waves at the edge of the Ag strips, with thicknesses in the range of micrometers, was carried out before the test in the subwavelength regime, with the help of a new improved transducer with metamaterial (MM) lenses. By simulation, a field snapshot was obtained in each sequence of geometry. The evanescent waves are emphasized in the plane XY, due to the scattering of the field on the edge of the strips. On the other hand, ZnO nanoparticles are investigated as a convenient high-efficiency biodetection material, where these structures were used as a biosensitive element to various enzymes (glucose, cholesterol, uric acid, and ascorbic acid). The obtained results demonstrate that the investigated structures based on ZnO nanostructures deposited on different supports are fast and sensitive for enzyme detection and can be successfully incorporated into a device as a biosensing element.

## 1. Introduction

The combined discipline of nanobiosensing is a result of how engineering sciences, physics, chemistry, and biology coincide at the nanometer scale. Nanomaterials obtained from metal oxides, shaped into nanoparticles, have been extensively used for their capacity as electrode biosensing elements to enhance the efficiencies of electrochemical biosensors [1]. Metal oxide nanoparticles can increase the efficiency of photochemical reactions and greatly improve the catalytic activity of enzymes to generate novel photoelectrochemical systems [2]. Due to the high electrical conductivity of the resulting size of the nanostructured metal oxides, the designed biosensors enhance the signal-to-noise ratio and the sensitivity by more than one order of magnitude compared with that observed in bulk material electrodes [3]. This improved performance was due to the rich enzyme loading and a better electrical communication capability of the nanomorphological structure and the active center of the biomolecules. The diverse electroanalytical techniques such as pulse and square-wave voltammetry, electrochemical impedance spectroscopy, and cyclic voltammetry were employed to analyze the electron transport phenomena such as diffusional mass transport, adsorption, chemisorption, charge transfer, chemical reaction, and convection [4]. The metal oxide nanoparticles’ modified electrochemical interfaces behave as a nanostructured biosensing element. Therefore, the electroanalytical limit of detection of a nanostructured element may be much lower than that of an analogous macrosized element because there is a bigger ratio between faradaic (the current generated by the reduction or oxidation of some chemical substance at an electrode) and capacitive currents. Due to the significant features of nanostructured metal oxide that show significant enhancements, much of the literature has reported that metal-oxide-modified interfaces have become biosensitive [2]. Nanostructured metal-oxide-based enzymatic biosensors have excellent prospects for interfacing biological recognition events with electronic signal transduction so as to design a new generation of bioelectronics devices with high sensitivity. Indeed, there has been substantial progress in the past decade on enzymatic electrochemical biosensors.

The biosensor is an analytical device formed by a biosensing matrix (receptors, nucleic acids, enzymes, antibodies, and microorganisms) and a transducer (electrochemical, piezoelectric, calorimetric, photometric, and acoustic/mechanical) that transforms biological features into measurable signals as shown in Figure 1 [1]. Enzymes play a major role in any kind of biosensor as they are the recognition unit responsible to deliver the information to the transducing device.

The hybridization of individual nanostructures can combine or even improve the properties of each component. Metal nanoparticles are typically characterized by their high surface activity, large specific surface area, good electron conductivity, strong biomolecule absorbability, and biocompatibility [5,6]. ZnO nanoparticles and nanostructured thin films have been studied in terms of their gas-sensing abilities and biosensing element [7,8] where it was shown that a rough surface with a larger surface area had a positive influence on the sensor response [9,10,11,12,13].

Metal strip gratings (MSGs) can act as filters or be used as conductive strips o micro-strips in specific applications such as microsensors, flexible printed circuits, etc. [14]. The interaction of electromagnetic fields with periodic metallic structures is a subject of study both theoretically and experimentally. The thin MSGs with silver features and subwavelength operating at 500 MHz and the formation of many evanescent modes in slits as well as anomalous modes was revealed by using metamaterial (MM) lensing and/or circular aperture diffraction with the subwavelength diameter [15,16]. The investigation of thick MSG with silver bands and a transverse magnetic-excited subwavelength feature polarized field, TMz, in the RF domain, proved the formation of a single evanescent mode in the slits, whose maximum was located in the strips, in the proximity of the lateral edge. This mode was detected experimentally using MM lenses with the investigated structures having great applicability potential as sensors and biosensors that work in the subwavelength regime from the perspective in which evanescent waves can appear in slits and the possibility of being a biosensitive element [12]. Relatively recent research has revealed that metallic structures with sub-wavelength features in the electromagnetic (EM) field at RF and microwave frequency range have potential for producing unusual EM response [17,18], creation of high impedance surface [19], have an effective plasmonic behavior [20], and high index of refraction and negative refractive index [21]. The grating structures constructed from subwavelength dimension components [22] can be used in the design of biosensors so that high sensitivity is achieved. The analyte particles penetrate the deposited biosensitivity material modifying the refractive index of the assembly [19,22]. The diffraction limits using a sensor with MM decrease until 300 nm instead of 100 μm in the case of conventional sensors and biosensors [23,24]. Extending the analysis to mesostructures, we focus on the study and obtainment gratings with Ag strips deposited on different substrates by vacuum thermal evaporation method at room temperatures; both samples were obtained.

This paper presents the results obtained at the EM nondestructive evaluation of strip gratings deposited on different substrates used as bioactive surfaces using the EM sensor with MM lens in order to improve the appearance of evanescent waves when the slits of gratings are filled with immobilized enzymes [22]. This paper examines the performance of the ZnO nanoparticles’ biosensing properties in order to establish their availability for use as convenient, high-reliability biosensing materials on various substrates (ITO/glass and SiO_2_/Si).

## 2. Materials and Methods

### 2.1. Preparation of Ag/ZnO Structures

A wide range of techniques to deposit thin films can be used, such as molecular beam epitaxy (MBE), single-source chemical vapor deposition, sol–gel, spray pyrolysis and Rf magnetron sputtering, and metal–organic chemical vapor deposition [24,25]. The materials used in this study are two structures that consist of Ag grids on zinc oxide deposited on two different substrates (Ag/ZnO/ITO/glass—#1 sample and Ag/ZnO/SiO_2_/Si—#2 samples) are shown in Figure 2. Deposition of Ag/ZnO structures on different substrates by thermal vacuum evaporation was performed on VUP5M equipment. The used substrates of 180 × 180 mm^2^ areas were cleaned using the well-known procedures: ITO/glass and glass substrates were cleaned with acetone, washed with deionized water and alcohol, and dried with hot air; SiO_2_/Si and nSi[100] substrates were cleaned with HF solutions, washed in deionized water, and dried with air. The deposition conditions were: pressure 10^−5^ Torr, evaporator–substrate distance 8 cm, deposition rate 0.05 nm/s, and a deposition time of 30 s. The Zn thin layer was subjected to an ambient heat treatment at 450 °C for 1 h and the ZnO phase was obtained. Ag gratings with different widths and thicknesses in the range of micrometers were obtained as follows: different masks were used, the evaporator–substrate distance was changed between 4–8 cm and the deposition time was varied between 2–15 min [26]. After nanostructured ZnO layers were obtained by depositing the metallic layer of Zn with the mask in Figure 2a and subjecting the sample to the thermal treatment of 450 °C, we followed whether the thickness of the silver film depends on the deposition rate by increasing the deposition time (Figure 2d). In order to do this, we used the mask in Figure 2b to deposit multiple sets of samples in which we increased the deposition time between 2–15 min (Figure 2c). We found that a deposition time t = 2 min results in a thin silver film with h = 900 nm thickness. Increasing the deposition time up t = 15 min, we obtained a thick silver film with h = 14 µm thickness.

Finally, the MSG structures in Figure 2c, marked in Table 1, (#1 sample [Ag/ZnO/ITO/glass], #2 sample [Ag/ZnO/SiO_2_/Si], #3 sample [Ag/ZnO/glass], and #4 sample [Ag/ZnO/nSi[100]]). In this paper, we studied #1 and #2 samples from the set of 5 because the metallic strip gratings were obtained on the samples having conductive traces made of silver with 14 µm thickness and 1.2 mm width deposited on ZnO with the distance between traces being 0.8 mm (Table 1).

### 2.2. Characterization of Ag/ZnO Structures

The morphologies and structural characterization of the as-grown ZnO structures were characterized by X-ray diffraction (XRD), scanning electron microscopy (SEM), and atomic force microscopy (AFM). The XRD patterns were made on a diffractometer using CuKα radiation (λ = 1.5406 Å) in order to identify the phase composition of the samples (Figure 3). The crystallinity and crystalline ZnO nanoparticles were observed by the X-ray diffraction patterns. The samples were analyzed in the range of 2θ = 5°–80° with a scanning angle rate of 0.02 and a 2 s/step count time. The sharp and narrow diffraction of the peaks demonstrated that all ZnO thin films were of good crystalline quality. The reflection peaks at (100), (002), (101), (103), and (112) were indicative of the hexagonal wurtzite ZnO nanostructure. XRD results demonstrate that there are two peaks in the pattern at 34.4° and 36.2°, which correspond with the (002) and (101) planes of ZnO, respectively. The XRD patterns showed significantly sharp peaks for Ag, ZnO, and ITO. Remarkably, the XRD results for both samples show diffraction peaks of silver at (111) and (200) planes at 2θ = 38.2° and 44.4°.

The weakness of the peaks is related to the thickness of the thin films. The peaks correspond directly to the hexagonal structure of the ZnO [27,28,29]. This is due the nature of the source material, and it is assumed that only nanoparticle migration from the source to the substrate takes place. All of the indexed peaks in the obtained pattern are well-matched with that of bulk ZnO, which confirms that the synthesized products are crystalline and possess a wurtzite hexagonal structure. No other peak related to impurities was detected in the pattern within the detection limit of the X-ray diffraction, which further confirms that the obtained products are pure ZnO.

Structural characterization was performed by scanning electron microscopy (SEM) analysis using a JEOL JSM 6390 instrument. SEM images show that the surfaces of the strips are free of inclusions and defects, making them suitable for radio frequency applications. From the SEM analysis, it was also highlighted that the obtained structures achieve the necessary quality for the study of evanescent waves as the penetration depth of the radio frequency waves is comparable to the thickness of the tape grating. SEM images confirm that the grown structures are crystalline ZnO nanoparticles synthesized in high density with uniform morphologies. SEM shows that the ZnO layer without the enzyme has a uniform film with a columnar structure of ZnO films (Figure 4a,b). The morphological characterization was performed by atomic force microscopy (AFM) with the AFM XE-100 instrument where AFM (Figure 5a,b) images of the ZnO layer with the enzyme show many globular structures [30,31,32], confirming immobilization of different enzymes.

The influence of nanoparticle size on the performance of investigating biosensors and nanoparticles with smaller size were established to be more suitable for enzyme immobilization [33]. Other studies reported on designing of biosensors based on the immobilization of different enzymes with nanostructure metal oxides, such as glucose oxidase, cholesterol oxidase, urease, hemoglobin, cytochrome C, tyrosinase, etc. [34]. Electrical contacting of redox enzymes with electrodes is a key process in the construction of third-generation enzyme electrodes. Although biosensing systems used a diversity of recognition elements, electrochemical detection mechanisms use preponderant enzymes. This is mostly due to their specific binding capabilities and biocatalytic activity. The electrochemical reactions can be utilized for construction of amperometric biosensors. The electrochemical biosensor is defined as a self-contained device that is capable of providing specific and quantitative or semi-quantitative analytical information using a biological element (in this case an enzyme), retained in direct spatial contact with an electrochemical transduction element [35]. The detecting mechanism for our glucose/cholesterol biosensors consist of the voltage applied across the two electrodes which causes a current to flow via electron tunneling through the potential barrier between the nanoparticles. The exhaustion region at the surface of the film, produced from a mixture of cholesterol oxidase/glucose oxidase and ZnO nanoparticles, is extended by the electrical field of electrons generated by the reaction between cholesterol oxidase/glucose oxidase and glucose/cholesterol. The ZnO nanostructures reoxidize by transferring the electron to the external circuit due to efficient electron transfer and a good redox property of the prepared nanostructures biomatrix. The increase in current with increasing concentration of glucose/cholesterol is attributed to the increase in the number of released electrons during oxidation of glucose/cholesterol.

## 3. Experimental Set-Up and Simulations

The investigation of the appearance of evanescent waves at the edge of Ag strips, with thicknesses in the range of micrometers, was carried out in the subwavelength regime, with the help of a new improved transducer with metamaterials lenses. The sensor based on the MM lens described in ref. [36,37,38] is coupled to Anritsu MS2028C VNA, improving the spatial resolution to obtain the EM images using Fourier optics [39] and the experimental set-ups presented in ref. [12,15] were used at a working frequency of 874 MHz (Figure 6). The finite-difference time-domain (FDTD) method was chosen as a simulation procedure with XFDTD software [40] because it involves a fine mesh to avoid numerical dispersion, establishes well-defined boundary conditions and an effective absorbing boundary conditions, and chooses proper excitation in space and time [41]. By simulation, a field snapshot can be obtained in each sequence of geometry. The evanescent waves are emphasized in the plane XY due to the scattering of the field on the edge of the strips. Figure 6a presents the result of simulation with XFDTD where the E_y_ component is displayed. In ref. [26], the behavior of the field with air in the slits is presented and it can be shown that for uric acid between the strips, the amplitude of the electric field has the same behavior as in [21] but the amplitude decreases due to the high electrical permittivity of uric acid. Thus, the symmetrical maxima appear in the middle of the slits, decreasing to the minimum value on the strips’ edges. Inside the strips, another pair of maxima appear, followed by a decrease to the middle of the strip.

During the measurements, the transducer is fixed and the samples are displaced in front of the transducer using an XY motorized stage, Newmark type, with the scanning step being 0.1 mm. The EM sensor with metamaterial lenses is connected to an Anritsu MS2028C VNA (Figure 6a), in the interval 850 MHz–900 MHz at 6 equidistant frequencies, with the optimal frequency of 874 MHz being confirmed. The distance between the screen with circular aperture and the surface to be examined (lift-off) is 20 ± 1 μm. Figure 7b shows that the dependency of the e.m.f. amplitude induced in the reception coil of the sensor correctly emphasizes the extremely thick conductive strips and eventual interruptions. The realized samples were deposited on different substrates, without and with enzymes, and were fixed on a support on a system that assures XY displacement—Newmark USA, controlled by PC through codes written in Matlab 2019b. The distance between the screen aperture and the surface to be examined was maintained at 20 μm ± 1 μm [16,36]. By simulation, a field snapshot was obtained in each sequence of geometry. The evanescent waves are emphasized in the plane XY, due to the scattering of the field on the edge of strips.

Considering the middle of the slits as a reference point, symmetrically to left and right, a 1 mm distance was scanned, and along the length of the strip, a 1 mm distance was scanned; thus, a surface of 1 × 2 mm^2^ is scanned, both on the sample without enzyme as well as the sample with enzymes. The EM field induced in the reception coil was measured; the measurement system represents the average of 20 measurements for each point in order to reduce the white noise effect, with the bandwidth of the analyzer being set to 10 Hz to diminish the noise level. The acquisition through the IEEE 488 interface and storage of data are made by the same PC. Considering the middle of the slits as a reference point, symmetrically to left and right, a 10 mm distance was scanned, along the length of the strip distance, so that a surface of a 2 × 2 mm^2^ structure was scanned in 10 µm steps in both directions, both on the sample without as well as the sample with enzymes. Scanning along the y direction corresponds to a period of the grid in the frequency range 850 MHz–900 MHz with a step of 1 MHz. Correlating the results obtained by measuring the amplitude of the EM field with the conditions obtained, it can be seen that the surface of the band becomes compact and dense and the distribution of the electric field along the band is approximately uniform (amplitude measured at each point represents the average of twenty different measurements) and evanescent waves can be observed. The spatial resolution of the system (distance between two distinctively visible points) was verified according to [42]. For gratings with features compatible with the value of the incident field, the Tez polarized wave acts at normal incidence for gratings [43]. Using the sensor described above, the results obtained at the scanning of 2 × 2 mm^2^ of both grating mesostructures with 10 μm steps in both directions at a frequency of 874 MHz are presented below.

## 4. Results and Discussion

Electrochemical biosensors are the most common biosensors, and are more efficient than conventional measurement techniques such as NMR spectroscopy, radioisotope tracing, and microfluorometric assays [44,45]. Potentiometric measurements usually lack high sensitivity because of the semi-logarithmic relationship between sensor output and analyte concentration [46,47]. Figure 8 shows the relation between the current and enzyme concentration for both mesostructures considered as biosensing materials. It can be observed that the current increases when the concentration and saturation appears at higher concentrations of enzymes. The inset is a calibration curve at λ = 0.6 m. The calibration curves were obtained in the range of 0.01–9.5 mM of enzymes and the current response showed a linear dynamic range of 0.04–4 mM.

The tested sensor configuration showed large dynamic ranges with an output response that was linear versus the concentration of the enzymes activity with sensitivity at a lower response current as shown in the inset in Figure 8a. To evaluate the performance of the studied biosensing materials, the parameter of selectivity, which is an important characteristic to describe the specificity towards the target ion in the presence of urea and ascorbic acid ions, was checked. The calibration shows the study of interferences versus the time traces line output response change with the time for both mesostructured biosensing materials. The photocurrent differences (ΔI) before and after the addition of enzymes is used to compare the photoelectrochemical response of both biorecognition samples: where I_0_, I are the photocurrents of both modified samples before and after the addition of cholesterol and glucose, respectively. The regression equation for the #2 sample is R = 0.999 at cholesterol and for the #1 sample it is R = 0.998 at glucose. Thus, the value of cholesterol biosensitivity for the #2 sample (1.94 µA/mM) is much better than the value of glucose biosensitivity for the #1 sample (1.33 µA/mM). As a comparison (Figure 8b), the photocurrent response of the ZnO nanoparticles with enzymes is higher compared with the sample deposited on SiO_2_ vs. ITO/glass support. The results showed that the favorite support is ZnO nanostructures for active surface biosensing, which indicates that the enhancement of the current response resulted from the cooperation of ZnO nanoparticles and enzyme. In checking possible interference from reducing agents such as urea and ascorbic acid, which are well-known to interfere with glucose and cholesterol measurement methods, their addition did not substantially change the signal. Adding 0.05 mM of urea and ascorbic acid to 0.5 mM glucose and cholesterol only generated some noise shown in Figure 8d. The morphology of the nanostructure significantly affects biosensing element properties where ZnO nanostructures have been investigated for application in enzyme biorecognition materials. Thus, the #1 sample exhibits good performance as a biosensing material for glucose as well as the #2 sample exhibits for cholesterol (Figure 8a). The response time is fast, which exhibits a high electron communication feature of the used thin ZnO films (Figure 8d).

To check the stability of the realized biosensors, the experiments were conducted over a long period to investigate the storage stability. As one result, the zinc oxide nanoparticles deposited onto the silicon oxide substrate have the highest biorecognition of all tested enzymes. This behavior can be attributed to the concentration of the surface states. It was found that the concentrations of the surface states are lower for the films deposited onto silicon than those for the films deposited onto glass substrates. It was concluded that the nature of the substrate is an important parameter which influences the enzymes’ biorecognition of the zinc oxide nanoparticles. The response time is also a key parameter defined as the time at which the photocurrent of the biorecognition elements reaches a saturation value upon exposure to enzymes.

In repeat experiments from the stored nanostructures biorecognition element, it was found that the realized biosensors did not show any significant decrease in biosensitivity for more than four weeks while being stored in an appropriate form when not in use. It was found that 98.5% of initial biorecognition values were retained up to four weeks and then gradually decreases. This might be due to the loss of the catalytic activity. The obtained results clearly suggest that the realized biosensor can be used for more than one month without any significant loss in biosensitivity.

## 5. Conclusions

The results showed that ZnO nanoparticles form an agreeable structure for enzyme immobilization, which exhibits good affinity, high sensitivity, and fast response for glucose/cholesterol detection. The research discusses the effectiveness of the EM sensor for experimentally highlighting the evanescent modes created by TEz polarized EM waves in the RF spectrum at a frequency of 874 MHz that become evident in mesostructures in the subwavelength regime. The presence of evanescent modes was noted in the case where enzymes fill the gaps between the grating strips, supporting the finding that the presence of an enzyme (such as glucose or cholesterol) in the slits increases the signal amplitude in the receiving coils as well as the photoelectrochemical response of both biorecognition samples. The obtained results demonstrated that the investigated structures based on ZnO nanostructures deposited on different supports are fast and sensitive for enzyme detection and can be incorporated into a biosensor. Glucose/cholesterol biorecognition offers the potential to fabricate high-performance biosensors for operation in air. While sample #2 shows a better response time for cholesterol with a biosensitivity of 1.94 µA/mM and R = 0.999, sample #1 has a better response time for glucose with a biosensitivity of 1.33 µA/mM and R = 0.998. The investigated biosensing materials exhibit good performance in terms of improved sensitivity, selectivity, reproducibility, stability, minimal interference, and quick response, making them suitable for the integration or external interface of a biosensing element with commercial devices. This adds the benefits of simplicity and low cost for enzymatic detection of substances with biomedical engineering applications. The results showed that the support of ZnO nanostructures stimulates the active surface by improving the response due to the immobilization of enzymes on ZnO nanoparticles. The researched structures can be used as biosensitive elements in the wireless monitoring of physiological parameters in biomedical engineering, environment, food industry, as well as in other cutting-edge domains. Based on ZnO nanostructures in a metallic MSG framework, a quick and effective method has been employed to biosense glucose and cholesterol.

## Figures and Tables

**Figure 1 micromachines-13-01997-f001:**
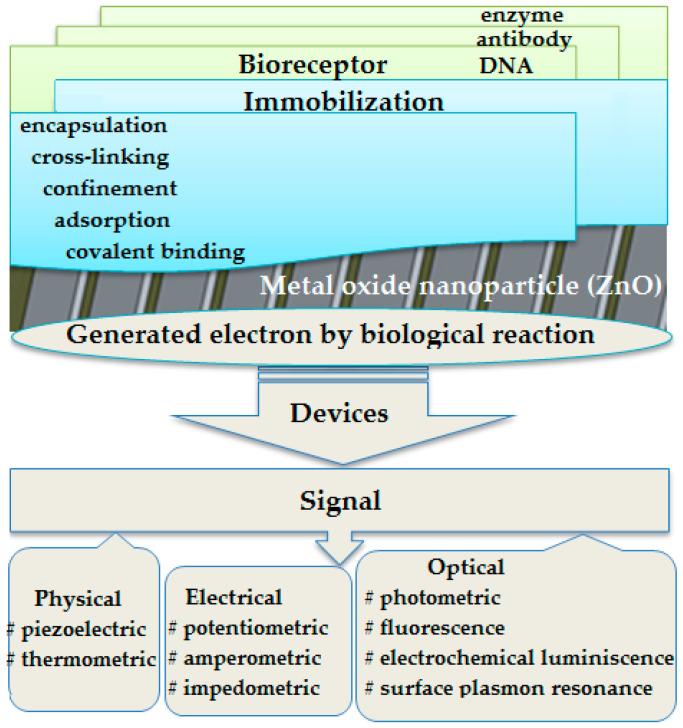
Working principles of a biosensing materials-based enzyme.

**Figure 2 micromachines-13-01997-f002:**
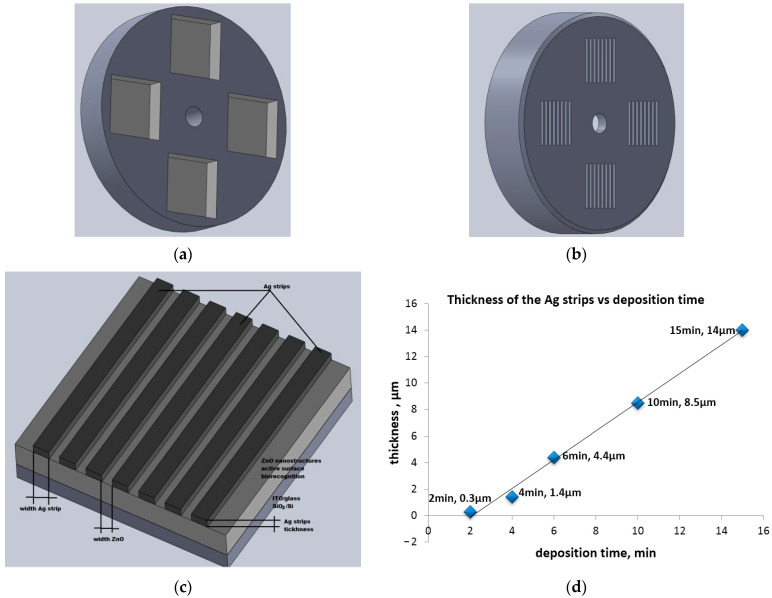
(**a**) Mask for Zn deposition; (**b**) mask for Ag strip deposition; (**c**) metallic structure type MSG; and (**d**) thickness vs. time.

**Figure 3 micromachines-13-01997-f003:**
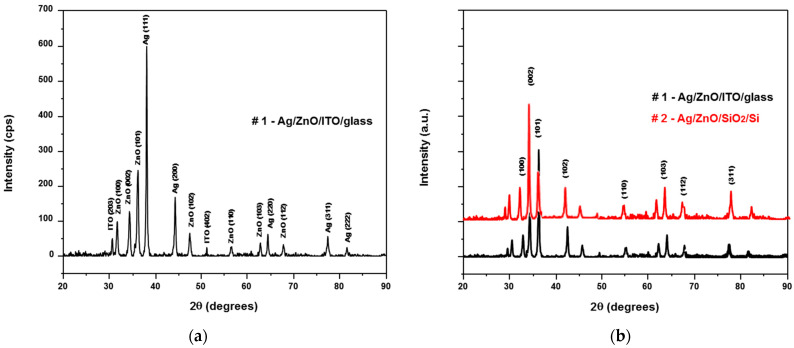
(**a**) XRD for #1 sample as grating nanostructures; and (**b**) XRD both ZnO thin film nanostructures.

**Figure 4 micromachines-13-01997-f004:**
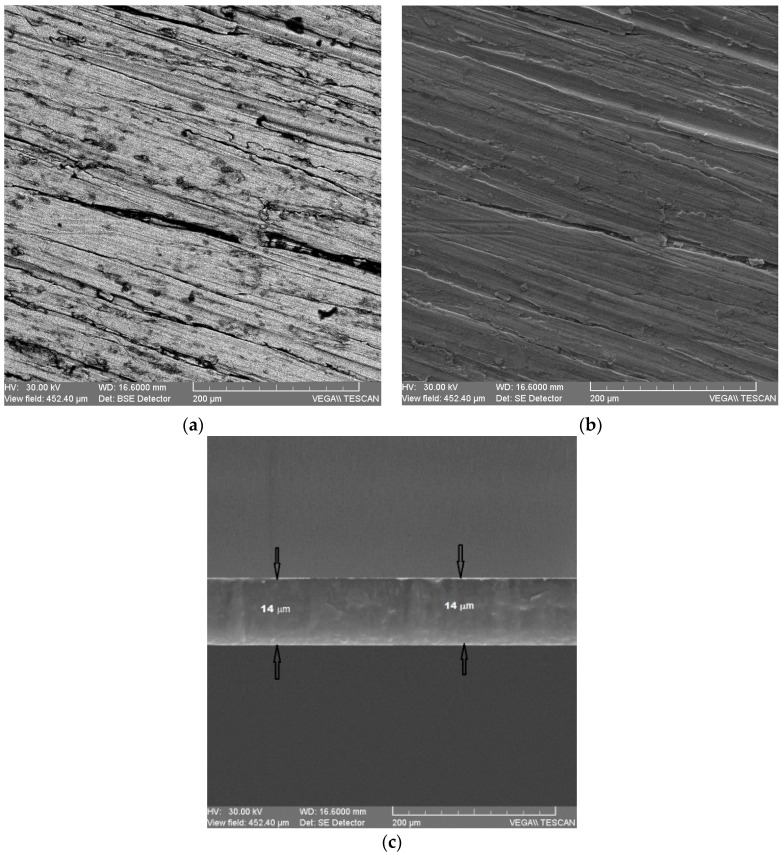
The SEM of ZnO thin films deposited on substrates without enzyme: (**a**) ITO/glass; (**b**) SiO_2_/Si; (**c**) SEM image with 14 μm thickness of Ag strip.

**Figure 5 micromachines-13-01997-f005:**
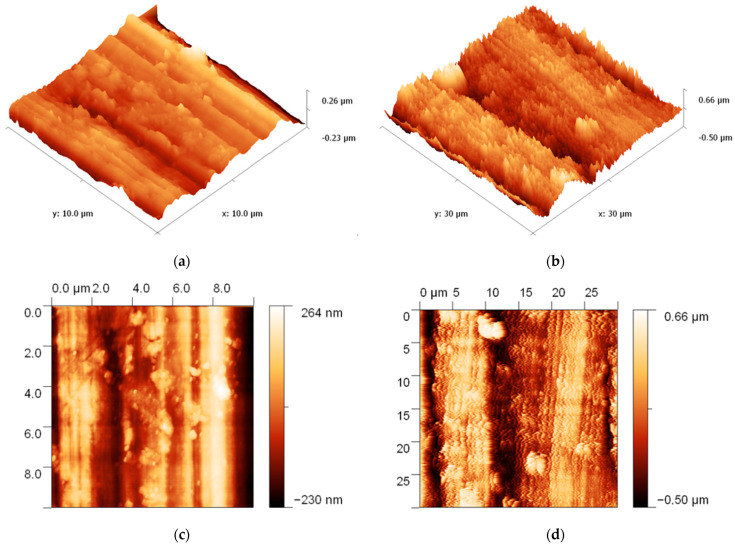
3D AFM images of: (**a**) #1 sample: glucose oxidase with immobilized enzyme glucose; (**b**) #2 sample: cholesterol oxidase with immobilized enzyme cholesterol; (**c**,**d**) 2D AFM images of #1 sample and #2 sample.

**Figure 6 micromachines-13-01997-f006:**
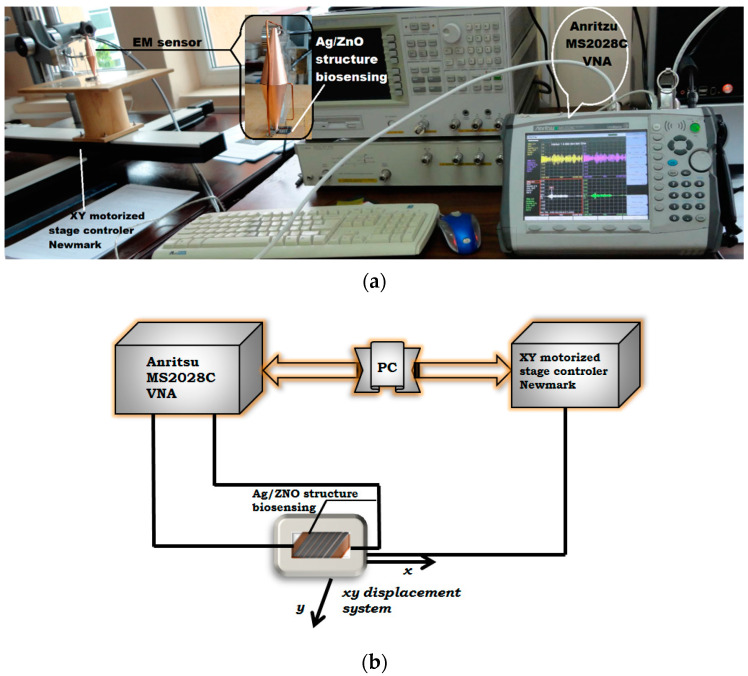
(**a**) experimental set-up; (**b**) block scheme.

**Figure 7 micromachines-13-01997-f007:**
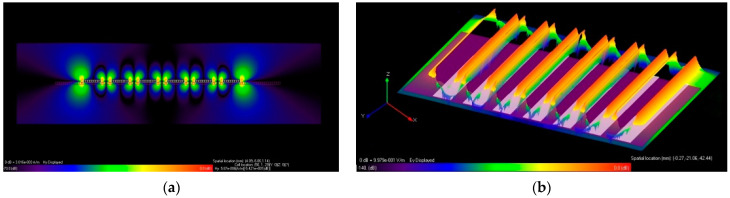
Numerical results for electric-field amplitude distribution near the strips; the field values are normalized to the amplitude of the incident field: (**a**) simulation with XFDTD where the Ey component and (**b**) dependency of the e.m.f. amplitude are induced in the reception coil of the sensor.

**Figure 8 micromachines-13-01997-f008:**
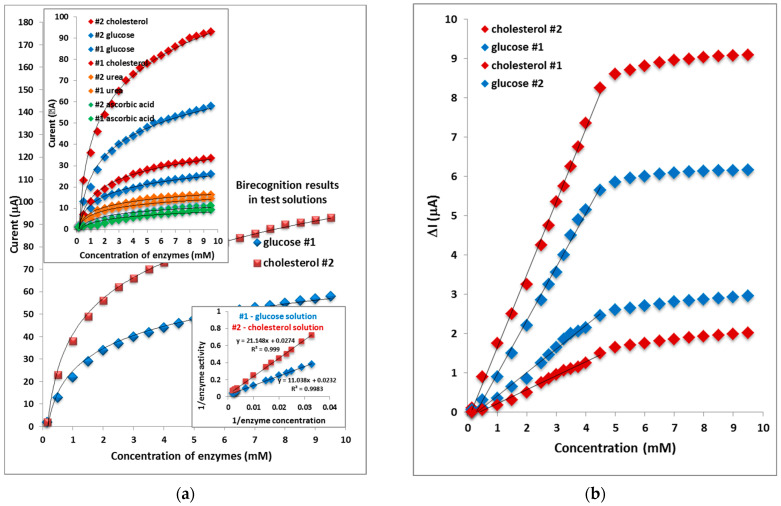
(**a**) Experimental results in test solutions: #1 and #2 samples; (**b**) calibration biorecognition samples with concentration of enzymes; (**c**) bar diagram of the concentration of enzymes in both samples; (**d**) current density response over time in the presence of increasing concentrations of enzymes.

**Table 1 micromachines-13-01997-t001:** The characteristics for the obtained metallic strip grating structures emphasizing the #1 and #2 samples from the set of 5 in the study.

No. Sets	Metallic Strip Grating Structures (MSG)	Deposited Time [min]	Width (Ag) [mm]	Width (ZnO) [mm]	Thickness Ag Strip [μm]
set I	#1 [Ag/ZnO/ITO/glass]	2	[0.9–1.2]	[0.6–0.8]	0.9
	#2 [Ag/ZnO/SiO_2_/Si]				
	#3 [Ag/ZnO/glass]				
	#4 [Ag/ZnO/nSi[100]]				
set 2	#1 [Ag/ZnO/ITO/glass]	4	[0.9–1.2]	[0.6–0.8]	1.4
	#2 [Ag/ZnO/SiO_2_/Si]				
	#3 [Ag/ZnO/glass]				
	#4 [Ag/ZnO/nSi[100]]				
set 3	#1 [Ag/ZnO/ITO/glass]	6	[0.9–1.2]	[0.6–0.8]	4.4
	#2 [Ag/ZnO/SiO_2_/Si]				
	#3 [Ag/ZnO/glass]				
	#4 [Ag/ZnO/nSi[100]]				
set 4	#1 [Ag/ZnO/ITO/glass]	10	[0.9–1.2]	[0.6–0.8]	8.5
	#2 [Ag/ZnO/SiO_2_/Si]				
	#3 [Ag/ZnO/glass]				
	#4 [Ag/ZnO/nSi[100]]				
set 5	**#1 [Ag/ZnO/ITO/glass]**	**15**	**[0.9–1.2]**	**[0.6–0.8]**	**14**
	**#2 [Ag/ZnO/SiO_2_/Si]**				
	#3 [Ag/ZnO/glass]				
	#4 [Ag/ZnO/nSi[100]]				

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
