# Peer review of "Metallic Structures Based on Zinc Oxide Film for Enzyme Biorecognition"

_micromachines, 2022, doi:10.3390/mi13111997_

Round 1

Reviewer 1 Report

The authors present the study of biomedical engineering applications based on ZnO nanoparticles structures.

The authors present their work in a fuzzy manner and the reader cannot understand clearly the intention, the experimental setup and the conclusions of the study.

- The Introduction should be more concise. 

- Materials and methods section presents information that should be moved into the Results section. There are no information about the materials used (supplier, purity, features useful for the designed experiment). Preparation of samples for different analytical techniques is not presented. The experimental setup for XRD, SEM, AFM is not presented. The biosensing system device was purchased or it was prepared? Please provide this information.

-figure 4 presents AFM images with glucose oxidase and cholesterol oxidase without other explanations. Somehow, we can conclude that is a device with enzyme immobilized on the metallic structure but there is no information about the experimental setup used to obtain the studied layers.

- page 4 - lines 139 - 146 (XRD results) - the presentation from the entire paragraph is unclear. Besides, you present information like "the weakness of the peaks is related to the thickness of the thin films".  In my opinion this type of information needs a solid reference.

- page 4 - line 148 - "columnar behavior" should be replaced by "columnar structure"

- Page 4 -line 150 - 151 - you should not compare SEM images from fig 3 with AFM images from fig 4

- page 5 and 6 - you present general information about the preparation of biosensing systems. 

- the experimental setup section is unclear presented.

- The Results and discussion section should be revised.

Reviewer 2 Report

Suggest reviewing its title as it doesn’t seem reflect on the contents.

Some grammatical error and/or typos break the flow of reading. For example, line 13 & 20 in the abstract, line 140 on page 4, 161 on page 5, and many more. Need to have a proper and consistent tense. Suggest a thorough proofread.

Faradaic current In line 49 on page 1? A grammatical flaw in the first sentence on page 2 interrupts communication.

It is a typo fir the scanning angle range in line 128 on page 3?

The authors claim that the AFM images shown in Figure 4 show many globular structures, which is not the case from my eyes. Further a reference is required how globular structures are the evidence of immobilization of different enzymes.

The first paragraph of Section 3 is not quite readable and required to be rewritten.

Not everyone knows what XFdtd is. Further, nothing has been discussed in terms of numerical analysis and others’ work using XFdtd, but it came out of nowhere. Have you introduced what the role of this simulation?

Again a division sign is used to indicate a range (line 224 on page 7).

How does simulation result go? What’s the role of it?

How many samples prepared and tested? Considering differences in results between two solutions, a certain level of statistical analysis is required to ensure the results carry significance.

Round 2

Reviewer 1 Report

Thanks to the authors for their reply.

Please recheck your manuscript for the typos.

Reviewer 2 Report

Thanks for addressing the previous comments.